# Functional and mutational landscapes of BRCA1 for homology-directed repair and therapy resistance

Rachel W Anantha[1,2†], Srilatha Simhadri[1,2,3†], Tzeh Keong Foo[1,2†], Susanna Miao[4], Jingmei Liu[1,2], Zhiyuan Shen[1,2], Shridar Ganesan[1,2,3], Bing Xia[1*]

[1]Rutgers Cancer Institute of New Jersey, Rutgers, The State University of New Jersey, New Brunswick, United States; [2]Department of Radiation Oncology, Rutgers, The State University of New Jersey, New Brunswick, United States; [3]Department of Medicine, Robert Wood Johnson Medical School, Rutgers, The State University of New Jersey, New Brunswick, United States; [4]Department of Genetics, School of Arts and Sciences, Rutgers, The State University of New Jersey, Piscataway, United States

**Abstract** BRCA1 plays a critical role in homology-directed repair (HDR) of DNA double strand breaks, and the repair defect of *BRCA1*-mutant cancer cells is being targeted with platinum drugs and poly (ADP-ribose) polymerase (PARP) inhibitors. We have employed relatively simple and sensitive assays to determine the function of BRCA1 variants or mutants in two HDR mechanisms, homologous recombination (HR) and single strand annealing (SSA), and in conferring resistance to cisplatin and olaparib in human cancer cells. Our results define the functionality of the top 22 patient-derived BRCA1 missense variants and the contribution of different domains of BRCA1 and its E3 ubiquitin ligase activity to HDR and drug resistance. Importantly, our results also demonstrate that the BRCA1-PALB2 interaction dictates the choice between HR and SSA. These studies establish functional and mutational landscapes of BRCA1 for HDR and therapy resistance, while revealing novel insights into BRCA1 regulatory mechanisms and HDR pathway choice.

*For correspondence: xiabi@cinj.rutgers.edu

†These authors contributed equally to this work

Competing interests: The authors declare that no competing interests exist.

## Introduction

Germline, heterozygous mutations in *BRCA1* confer high risk of breast and ovarian cancer development in an autosomal dominant fashion (*Couch et al., 2014*; *Fackenthal and Olopade, 2007*). BRCA1 has been implicated in numerous cellular processes including DNA repair, cell cycle checkpoints, centrosome duplication, and transcriptional regulation, etc. (*Deng, 2006*; *Mullan et al., 2006*; *Roy et al., 2012*). Ever since BRCA1 was found to localize to discrete nuclear foci and colocalize with the recombination enzyme RAD51 (*Scully et al., 1997*), its function in homologous recombination (HR)-based repair of DNA double strand breaks (DSBs) has been a subject of intense study (*Moynahan and Jasin, 2010*). Tumors arising from *BRCA1* mutation carriers usually show loss of the wild-type (wt) allele, which renders tumor cells biallelically null for the gene. It is generally believed that genome instability resulting from the DNA repair defect following the loss of BRCA1 is a driver of tumor development (*Li and Greenberg, 2012*; *Venkitaraman, 2014*). Importantly, the very DNA repair defect that leads to tumor development is also an 'Achilles' Heel' of the resulting tumor cells, which can be selectively killed by suitable DNA-damaging agents that target HR defect, such as platinum drugs and poly (ADP-ribose) polymerase (PARP) inhibitors (*Lord and Ashworth, 2016*).

The human *BRCA1* gene consists of 24 exons encoding a large polypeptide of 1863 amino acid residues. BRCA1 contains a RING domain at the N terminus and a tandem BRCT domain at the C

**eLife digest** Genes are the instruction manuals of life and contain the information needed to build the proteins that keep cells alive. Over time, genes can accumulate errors or mutations and eventually become faulty, which can lead to diseases like cancer. Sometimes mutations can be passed on through generations and increase the chances of getting cancer. The *BRCA1* gene, for example, provides instructions for making a protein that helps to repair or remove damaged DNA and stops cells from growing uncontrollably. When the *BRCA1* gene becomes faulty, cells could continue to grow with damaged DNA. This makes it more likely for cancer to develop, especially breast cancer and ovarian cancer.

However, not all changes in *BRCA1* gene cause the protein to become faulty or lead to cancer. In fact, about 30% of *BRCA1* gene changes identified by genetic tests are referred to as 'variants of uncertain clinical significance', meaning that it is not clear if these variants are indeed mutations that could affect the clinical outcome of the people that carry them. Software predictions based largely on patient data have categorized many of these variants as not cancer-causing, but the majority still need to be experimentally tested and confirmed. Many studies have tried to determine the effect of selected variants on the BRCA1 protein, but a complete picture remains lacking.

Now, Anantha et al. have tested the top 22 common variants in the *BRCA1* gene, some of which had known effects and some did not. The study tested how these variants affect the ability of the protein to repair damaged DNA and the efficacy of chemotherapies targeting cancer cells with a DNA repair defect. The experiments revealed that three specific parts of the protein must remain intact in order for the protein to carry out this activity, i.e. mutations that affect these three areas are likely to cause cancer and also make cancer cells vulnerable to these chemotherapies. Anantha et al. also generated a series of 10 artificially shortened BRCA1 proteins, each missing a specific part, to determine the possible effects of other variants in those missing parts.

Together the findings reveal previously unknown effects of certain variants that are commonly seen in cancer patients as well new insights into how the BRCA1 protein repairs DNA. The next step will be to assess rarer variants where little data is available. A better understanding of how these variants affect DNA repair and drug response will help to improve the genetic counseling and treatment of patients with breast cancer and ovarian cancer.

---

terminus (*Figure 1A*). Two nuclear localization signals (NLSs) facilitate the localization of BRCA1 primarily to the nucleus (*Chen et al., 1996*), whereas a nuclear export signal (NES) can mediate its cytoplasmic export (*Rodríguez and Henderson, 2000*). The RING domain of BRCA1 binds to a similar domain in its close partner BARD1 (*Wu et al., 1996*), leading to the formation of a stoichiometric complex that possesses substantial ubiquitin E3 ligase activity (*Hashizume et al., 2001*; *Ruffner et al., 2001*). At the same time, binding of BARD1 to BRCA1 shields the NES and helps retain BRCA1 in the nucleus (*Fabbro et al., 2002*). The BRCT domain directly binds, in a phosphorylation-dependent manner, to at least three other proteins, namely BRIP1, CtIP and Abraxas, all of which have function in DNA repair (*Huen et al., 2010*; *Jiang and Greenberg, 2015*). Additionally, BRCA1 contains a highly conserved coiled-coil (CC) motif that directly binds PALB2, the partner and localizer of BRCA2 (*Xia et al., 2006*), which links BRCA1 and BRCA2 in the HR pathway (*Sy et al., 2009*; *Zhang et al., 2009a*, *2009b*).

The Breast Cancer Information Core (BIC) database was the first major public database into which patient-derived mutations and variants were deposited. It contains a total of 15,311 entries of *BRCA1* sequence alterations, among which 6133 (40%) are frameshift or protein-truncating mutations, which can generally be classified as pathogenic. Importantly, 4577 (30%) are missense variants, most of which are considered to be of 'unknown clinical importance', or commonly known as variants of unknown (or uncertain) clinical significance (VUSs), the interpretation of which remains a challenge (*Couch et al., 2014*). Recently, more data have been deposited into the ClinVar database, which now includes all BIC cases along with ones from multiple other sources. Based largely on genetic and epidemiological evidence, a large number of VUSs has been designated by ClinVar as 'benign'. However, the majority of the variants have not been functionally characterized.

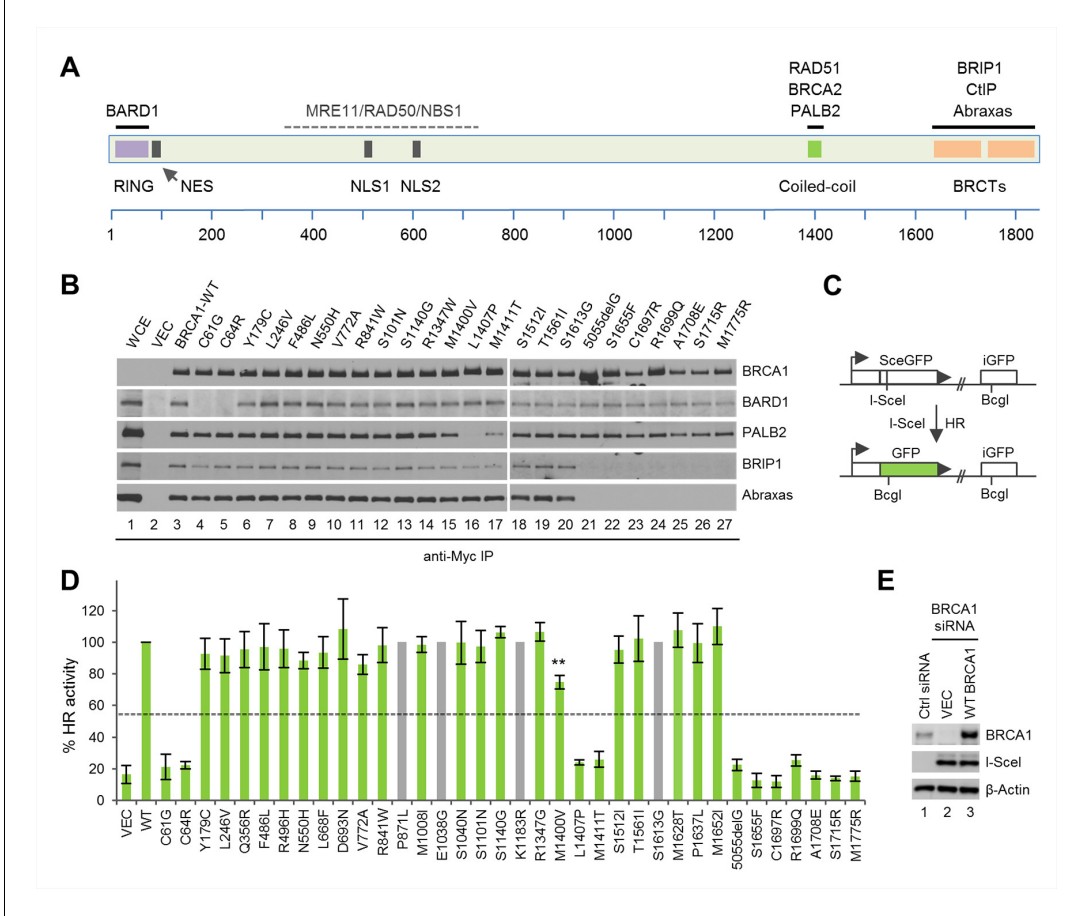

**Figure 1.** Sequence alterations generated in BRCA1 and their effects on protein-protein interactions and HR. (A) Domain structure of BRCA1 and the binding sites for its interacting partners. NES, nuclear export signal; NLS, nuclear localization signal. (B) Effects of the BRCA1 variants on the binding of BARD1, PALB2, BRIP1 and Abraxas. The 3xMyc-tagged BRCA1 proteins were transiently expressed in 293T cells and IPed with an anti-Myc antibody. WCE, whole cell extract. (C) Schematic of the HR reporter assay. The DR-GFP reporter contains two defective copies of the GFP gene, one disrupted by an I-SceI site and the other lacking a promoter. I-SceI cutting of the first copy generates a DSB, and repair by HR with the second copy as a template leads to restoration of a functional GFP gene. (D) HR activities of the variants relative to the wt BRCA1 protein. Data shown are the means from two to seven independent experiments for each variant or mutant. Error bars represent standard deviations (SDs). The grey bars indicate variants that are among the top 20 missense variants but are already present in BRCA1 cDNAs obtained from three independent sources. The calculated cutoff threshold is indicated by horizontal lines. **p<0.01. See *Figure 1—source data 1* for details. (E) Levels of BRCA1 protein following knockdown and re-expression. Cells treated with a control siRNA (NSC1) were used as a control for the endogenous protein abundance.

The following source data and figure supplements are available for figure 1:

**Source data 1.** HR activities of the BRCA1 variants and mutants analzyed in this study.

**Figure supplement 1.** Capacity of the BRCA1 variants to bind BARD1, PALB2, CtIP and BRIP1.

**Figure supplement 2.** Expression levels of BRCA1 BRCT missense and truncating mutants in U2OS/DR-GFP cells first depleted of the endogenous BRCA1 and then transfected with cDNA expressing the mutants.

A number of studies have been conducted to assess the functions of select *BRCA1* VUS in transcription, DNA repair, and in supporting the viability of mouse embryonic stem (ES) cells (*Bouwman et al., 2013*; *Chang et al., 2009*; *Millot et al., 2012*; *Ransburgh et al., 2010*; *Towler et al., 2013*). However, data on the impact of VUS on drug sensitivity have only begun to emerge. In particular, a recent study used a recombinase-mediated cassette exchange (RMCE)

approach to profile 74 patient-derived missense variants for their activities in conferring resistance to cisplatin and a PARP inhibitor in mouse ES cells, and it found that all functionally deleterious variants were confined to the RING and BRCT domains (*Bouwman et al., 2013*).

## Results

### The HR activity of BRCA1 requires intact RING, CC and BRCT domains

To establish an initial functional landscape of BRCA1 mutations for HR-mediated DSB repair, we constructed 32-patient-derived missense variants along the entire length of the protein (*Table 1*). The list included 11 variants with known or partially known functional consequences: two in the RING domain (C61G and C64R) that disrupt BARD1 binding, three in the CC motif (M1400V, L1407P and M1411T) that affect PALB2 binding and six in the BRCT domain (S1655F, C1697R, R1699Q, A1708E, S1715R and M1775R) that abrogate BRCA1 interaction with Abraxas, BRIP1 and CtIP (*Figure 1B* and *Figure 1—figure supplement 1*). Also included were the top 20-patient-derived missense variants in the BIC database (21 in total due to a tie at the 20th place) and the top 22 in the ClinVar database (*Table 1*). Additionally, we included four other VUSs, L668F, S1101N, S1140G and T1561I, which have now been recorded between 16 and 40 times in ClinVar. As a control for the BRCT missense variants, we also generated a patient-derived frameshift mutation (5055delG or p.Val1646-Serfs) that truncates the protein immediately before the BRCT domain.

Notably, our sequencing analysis revealed P871L, E1038G, K1183R and S1613G in the 'wild-type' BRCA1 cDNAs from multiple independent sources. Indeed, they are considered as neutral/benign by both BIC and ClinVar. Among the rest of the top 22 variants in ClinVar, C61G and A1708E are considered pathogenic, and the rest are all considered benign largely based on genetic or epidemiological evidence. Similarly, L668F, S1101N, S1140G and T1561I are now also considered as benign. Note that this study was initiated in 2010 largely based on BIC, in which most of the above are now still listed as VUS.

To measure the activity of BRCA1 variants in HR, we re-developed a previously described HR assay based on a 'protein replacement' strategy (*Ransburgh et al., 2010*; *Sy et al., 2009*; *Xia et al., 2006*) (*Figure 1C*). In this assay, the endogenous BRCA1 was first depleted in U2OS/DR-GFP cells (*Nakanishi et al., 2005*; *Xia et al., 2006*) using an siRNA targeting the 3'-UTR of the gene, and various BRCA1 proteins were then expressed from cDNAs (lacking the 3'-UTR) co-transfected with a plasmid expressing I-SceI, which cleaves the reporter to induce DSB formation and subsequent repair. To ensure uniformity of BRCA1 knockdown, siRNA transfections were performed in 10 cm plates and the cells were harvested, mixed and reseeded into six-well plates before the second transfection. The averaged expression levels of exogenous BRCA1 was three to five times that of the endogenous protein (*Figure 1E*), suggesting a significant degree of overexpression. However, the variants were expressed at levels comparable to that of the wt BRCA1 (not shown), with the exception being the BRCT point mutants, among which all but R1699Q showed much lower expression levels (*Figure 1—figure supplement 2*).

Under the setting used, C61G and C64R reduced BRCA1 HR activity by ~5-fold to a level slightly above the vector control (*Figure 1D* and *Table 2*). Similarly, a profound impact on HR was observed with mutations in the BRCT domain. Consistent with a previous report (*Sy et al., 2009*), variants L1407P and M1411T, which abrogate or substantially impair PALB2 binding, reduced HR by ~4.5-fold, whereas M1400V, which moderately impairs PALB2 binding (*Figure 1B*, lane 15), caused a ~ 25% decrease in the apparent HR activity. In contrast, all variants that are neutral for protein-protein interaction at the RING, coiled-coil and BRCT domains showed greater than 85% of the HR activity of the wt protein. Using variants that have been functionally characterized previously as controls, our assay was found to be both sensitive and specific, and the cutoff threshold of the assay was calculated to be 54.3–54.7%, which would define M1400V and all the protein-protein interaction neutral variants as being functionally benign. However, on an individual basis, the modest decrease of HR activity in the case of M1400V is statistically significant.

### The BRCA1-PALB2-BRCA2 interactions prevent RAD52-dependent SSA

Given the proposed role of BRCA1 on DSB end resection (*Bunting et al., 2010*), the activities of all the above variants in SSA, a deletion-causing repair pathway that also utilizes resected DNA ends,

**Table 1.** BIC database and ClinVar reports of patient-derived BRCA1 missense variants characterized in this study.

| HGVS cDNA | BIC designation | BIC entry count | BIC clinical importance | ClinVar individuals | ClinVar designation | Align-GVGD grade | IARC classification |
|---|---|---|---|---|---|---|---|
| c.4837A>G | S1613G | 248 | No | 1319 | Benign | C0 | 1 |
| c.181T>G | C61G | 239 | Yes | 323 | Pathogenic | C65 | 5 |
| c.2612C>T | P871L | 211 | No | 1348 | Benign | C0 | 1 |
| c.3113A>G | E1038G | 182 | No | 1237 | Benign | C0 | 1 |
| c.3548A>G | K1183R | 164 | No | 1215 | Benign | C0 | 1 |
| c.4039A>G | R1347G | 161 | Unknown | 712 | Benign | C0 | 1 |
| c.1067A>G | Q356R | 155 | Unknown | 315 | Benign | C0 | 1 |
| c.3024G>A | M1008I | 139 | Unknown | 146 | Benign | C0 | 1 |
| c.2521C>T | R841W | 119 | Unknown | 128 | Benign | C15 | 1 |
| c.4883T>C | M1628T | 96 | Unknown | 491 | Benign | C0 | 1 |
| c.1487G>A | R496H | 86 | Unknown | 90 | Benign | C0 | 1 |
| c.736T>G | L246V | 80 | Unknown | 83 | Benign | C0 | 1 |
| c.3119G>A | S1040N | 68 | Unknown | 141 | Benign | C0 | 1 |
| c.4910C>T | P1637L | 67 | Unknown | 76 | Benign | C0 | 1 |
| c.2077G>A | D693N | 65 | No | 229 | Benign | C0 | 1 |
| c.4956G>A | M1652I | 61 | Unknown | 94 | Benign | C0 | 1 |
| c.2315T>C | V772A | 60 | Unknown | 62 | Benign | C0 | 1 |
| c.4535G>T | S1512I | 58 | No | 72 | Benign | C0 | 1 |
| c.1456T>C | F486L | 56 | Unknown | 58 | Benign | C0 | 1 |
| c.536A>G | Y179C | 54 | Unknown | 59 | Benign | C35 | 1 |
| c.1648A>C | N550H | 54 | Unknown | 59 | Benign | C0 | 1 |
| c.5123C>A | A1708E | 46 | Yes | 80 | Pathogenic | C65 | 5 |
| c.5324T>G | M1775R | 31 | Unknown | 43 | Pathogenic | C45 | 5 |
| c.3418A>G | S1140G | 29 | Unknown | 40 | Benign | C0 | 1 |
| c.4682C>T | T1561I | 26 | Unknown | 33 | Conflicting | C0 | N/A |
| c.2002C>T | L668F | 25 | Unknown | 28 | Benign | C0 | 1 |
| c.3302G>A | S1101N | 14 | Unknown | 16 | Benign | C0 | 1 |
| c.190T>C | C64R | 12 | Unknown | 15 | Conflicting | C65 | N/A |
| c.5096G>A | R1699Q | 11 | Unknown | 22 | Conflicting | C35 | 5 |
| c.5145C>G | S1715R[#] | 5 | Unknown | 5 | Pathogenic | C65 | 5 |
| c.4964C>T | S1655F | 3 | Unknown | 4 | Conflicting | C25 | N/A |
| c.5098T>C | C1697R | 3 | Unknown | 7 | Conflicting | C65 | N/A |
| c.4198A>G | M1400V | 1 | Unknown | 1 | Uncertain | C0 | N/A |
| c.4220T>C | L1407P | 1 | Unknown | 1 | Uncertain | C65 | N/A |
| c.4232T>C | M1411T | 1 | Unknown | 1 | Uncertain | C65 | N/A |
| 5055delG | V1646Sfs | 5 | Yes | 6 | Pathogenic | - | - |

Align-GVGD grade: C0 to C65 denote increasing likelihood of a variant to cause damage (to protein function). IARC (ENIGMA) classification: 5, Definitely pathogenic; 4, Likely pathogenic; 3, Uncertain, 2, Likely not pathogenic or of little clinical significance; 1, Not pathogenic or of no clinical significance. # c.5145C>G, c.5145C>A and c.5143A>C have all been reported to generate BRCA1-S1715R. Data shown are up to date as of March 21, 2017.

**Table 2.** Comparison of results on HR activity and drug response of the BRCA1 variants analyzed in this study obtained from previous and this studies.

| | | HR activity (%) | | | | | | Cisplatin response | | Olaparib response | |
|---|---|---|---|---|---|---|---|---|---|---|---|
| | Domain | Sy et al., 2009 | Ransburgh et al. (2010) | Towler et al. (2013) | Bouwman et al. (2013) | Lu et al., 2015 | This study | Bouwman et al. (2013) | This study | Bouwman et al. (2013) | This study |
| Vector | | N/A | ~10 | ~9 | ~20 | ~18 | 17.3 | S | S | S | S |
| WT | | 98 | 100 | 100 | 100 | 100 | 100 | R | R | R | R |
| C61G | RING | - | ~17 | - | - | 23.6 | 21.2 | S | S | - | S |
| C64R | RING | - | - | - | - | - | 22.3 | - | S | - | S |
| Y179C | | - | - | ~95 | - | 157.1 | 92.7 | - | R | - | R |
| L246V | | - | - | - | - | - | 91.5 | R | R | - | R |
| Q356R | | - | - | - | - | - | 95.5 | - | R | - | R |
| F486L | | - | - | - | - | 160 | 95.8 | - | R | - | R |
| R496H | | - | - | - | - | - | 95.9 | - | R | - | R |
| N550H | | - | - | - | - | 90.8 | 88.4 | - | R | - | R |
| L668F | | - | - | - | - | 96.8 | 93.6 | R | R | - | R |
| D693N | | - | - | - | - | - | 111.5 | R | R | - | R |
| V772A | | - | - | - | - | 110.4 | 84.5 | - | R | - | R |
| R841W | | - | - | - | - | - | 98.2 | - | R | - | R |
| P871L | | - | - | - | - | - | wt | - | wt | - | wt |
| M1008I | | - | - | - | - | - | 99.2 | R | R | - | R |
| E1038G | | - | - | - | - | - | wt | - | wt | - | wt |
| S1040N | | - | - | - | - | - | 98.0 | - | R | - | R |
| S1101N | | - | - | - | - | - | 97.3 | R | R | - | R |
| S1140G | | - | - | - | - | - | 106.2 | R | R | - | R |
| K1183R | | - | - | - | - | - | wt | - | wt | - | wt |
| R1347G | | - | - | - | - | - | 106.5 | - | R | - | R |
| M1400V | CC | 56 | - | - | - | - | 74.8 | R | S | R | S |
| L1407P | CC | 24 | - | - | - | - | 24 | S | S | S | S |
| M1411T | CC | 25 | - | - | - | - | 26 | R | S | R | S |
| S1512I | | - | - | - | - | - | 95.3 | - | R | - | R |
| T1561I | | - | - | - | - | 133.2 | 102.3 | - | R | - | R |
| S1613G | | - | - | - | - | - | wt | - | wt | - | wt |
| M1628T | | - | - | - | - | 107 | 107.6 | R | R | - | R |
| P1637L | | - | - | - | - | 98.8 | 99.5 | - | R | - | R |
| 5055△G | BRCT | - | - | - | - | - | 22.5 | - | S | - | S |
| M1652I | BRCT | - | - | - | - | - | 106.2 | R | R | - | R |
| S1655F | BRCT | - | - | - | ~30 | - | 8.4 | S | S | - | S |
| C1697R | BRCT | - | - | - | - | - | 8.0 | - | S | - | S |
| R1699Q | BRCT | - | - | - | ~45 | - | 16.9 | S | S | S | S |
| A1708E | BRCT | - | - | - | - | - | 10.7 | - | S | - | S |
| S1715R | BRCT | - | - | - | - | - | 9.3 | - | S | - | S |
| M1775R | BRCT | 36 | - | ~6 | - | - | 10.2 | - | S | - | S |

R, resistant; S, sensitive. See **Figure 1—source data 1**, **Figure 2—source data 1** and **Figure 3—source data 1** for details.

were measured. For this purpose, we used a similar protein replacement strategy with another U2OS-based reporter cell line containing an integrated SA-GFP reporter (*Stark et al., 2004*). Consistent with a previous report that *Brca1*-deficient mouse cells show reduced SSA (*Stark et al., 2004*), depletion of BRCA1 in the human cells resulted in a reduction of the (already low) basal SSA activity (*Figure 2A*), and re-expression of wt BRCA1 restored the SSA activity (*Figure 2B*). As reported previously that pathogenic missense mutations in the RING domain can cause defects in both HR and SSA (*Towler et al., 2013*), we found that the BARD1-binding mutants (C61G and C64R) showed greatly reduced SSA activities (Figure 2B). At the same time, all the BRCT mutants were completely defective. In contrast, the three mutations that impact PALB2 binding all resulted in increased SSA activity. In particular, the L1407P and M1411T mutations elevated SSA by ~2.5-fold. This finding suggests that BRCA1 interactions with BARD1 and BRCT-binding partners are both required for its function in the resection step of the DSB repair, whereas the BRCA1-PALB2 interaction functions downstream to promote HR and reduce SSA.

It has been reported that loss of BRCA2 increases SSA in mouse cells (*Stark et al., 2004*; *Tutt et al., 2001*), and a recent report showed that some *PALB2* heterozygous human lymphoblastoid cell lines have increased SSA activity (*Obermeier et al., 2016*). To better understand the role of PALB2 and BRCA2 in this process, we depleted the two proteins in parallel in the U2OS/SA-GFP reporter cells and measured the effect. As shown in *Figure 2C*, depletion of either protein caused substantial upregulation of SSA, with BRCA2 siRNAs stimulating SSA by 1.8–5.4 fold and PALB2 siRNAs 5.6–7.2 fold. Initially, two siRNAs for each gene were used (#1949 and #9025 for BRCA2 and #1493 and #2693 for PALB2). The number of BRCA2 siRNAs were later increased to six due to the observation that siRNAs #1949 and #9025 showed a large difference in the fold change of SSA they induced. With the exception of siRNA #11170, all BRCA2 siRNAs showed similar and effective knockdown. Although we cannot explain why siRNA #1949 stood out as the sole outlier in terms of SSA induction, we did notice that this siRNA had the least negative effect on cell morphology and growth rate (data not shown). The level of RAD51 was reduced by all BRCA2 siRNAs, which could be explained by potentially reduced stability in the absence of BRCA2, a 'carrier' of RAD51 that can simultaneously bind six molecules of the latter (*Jensen et al., 2010*; *Liu et al., 2010*) and thus could potentially serve as a stabilizer as RAD51. However, there is no correlation between the levels of RAD51 reduction and SSA increase among all siRNAs used (*Figure 2C*), suggesting that the partial loss of RAD51 is not a significant cause of SSA upregulation.

RAD52 has been shown to possess strand annealing activity in vitro (*Mortensen et al., 1996*) and to promote SSA in vivo (*Stark et al., 2004*). To test if RAD52 is required for the increased SSA in the absence of BRCA2 and PALB2, we silenced its expression, either alone or in combination with BRCA2 and PALB2, and measured the effect on SSA efficiency. As shown in *Figure 2D*, strong depletion of RAD52 (by siRNA #1972) reduced basal SSA, whereas relatively mild depletion of RAD52 (by siRNA #2569) showed little effect. Strong RAD52 depletion greatly impeded the increase of SSA elicited by both BRCA2 and PALB2 loss, while the relatively weaker depletion of RAD52 also reduced the upregulation of SSA upon loss of either BRCA2 or PALB2, albeit to lesser extents.

To further elucidate the role of PALB2 in the DSB repair pathway choice, we employed point mutants that are defective for BRCA1 binding (L21A and L35A) or BRCA2 binding (A1025R) (*Figure 2D*) (*Oliver et al., 2009*; *Sy et al., 2009*). As expected, and in contrast to wt PALB2, which effectively restored HR in reporter cells depleted of the endogenous protein, the BRCA1- and BRCA2-binding mutants all showed dramatically reduced HR activity (*Figure 2E*). Re-expression of wt PALB2 in the SA-GFP reporter cells depleted of the endogenous PALB2 significantly suppressed the greatly enhanced SSA activity, whereas neither the BRCA1-binding mutants nor the BRCA2-binding mutant showed such ability (*Figure 2F*). Very recently, we reported the first patient-derived missense pathogenic mutation in PALB2, L35P, that disrupts the binding of BRCA1 (*Foo et al., 2017*). In a way virtually identical to L35A, this mutant also failed to suppress SSA in the above setting (data not shown). The suppression by wt PALB2 was moderate, which could be due to that the depletion of PALB2 might have caused certain secondary effects that cannot be immediately restored upon sudden re-expression of the protein. Taken together, our results establish that PALB2 controls the HR/SSA pathway choice following resection and that the interactions between PALB2 with both BRCA1 and BRCA2 are required for this function.

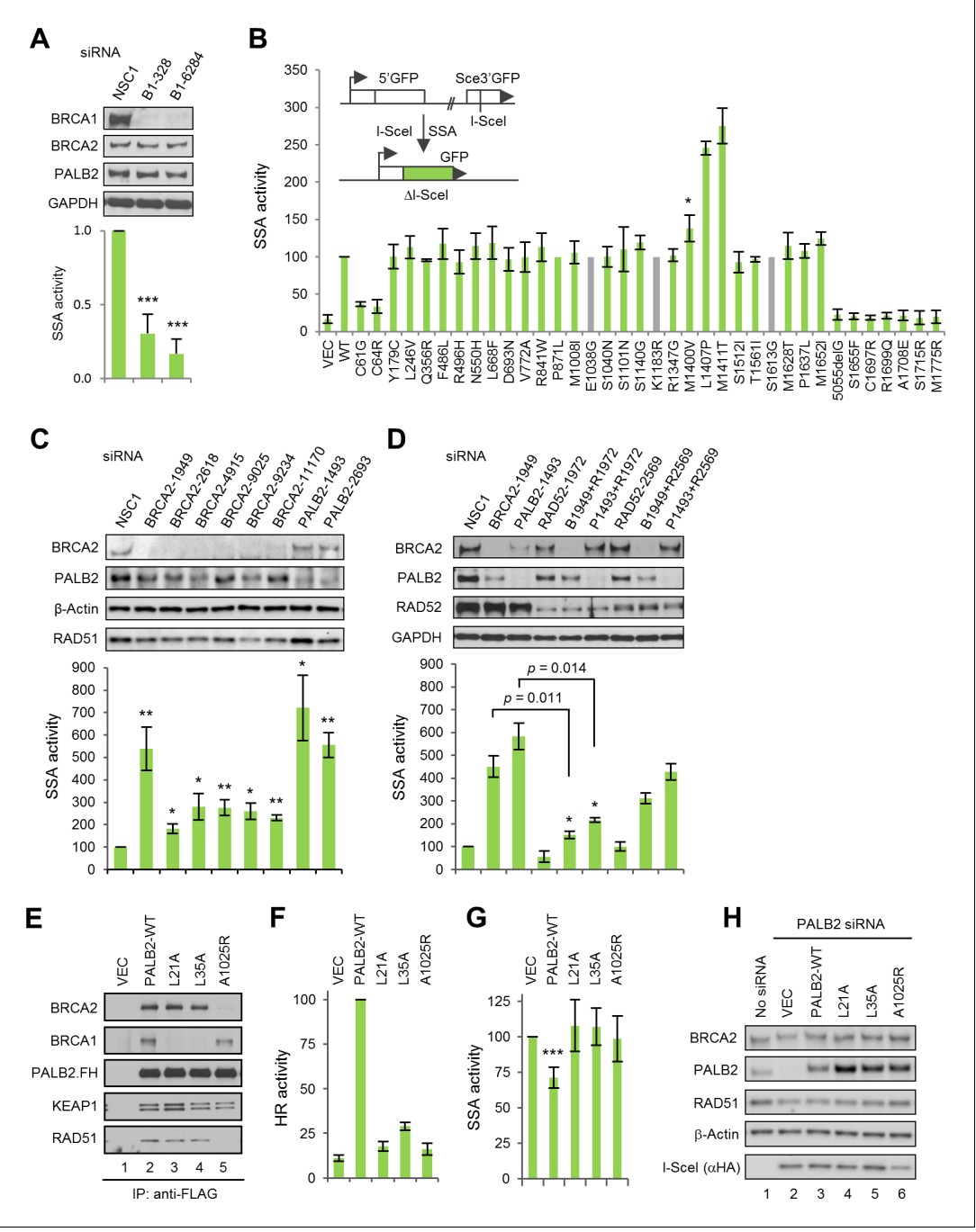

**Figure 2.** The BRCA1-PALB2 interaction suppresses SSA. (**A**) Effect of BRCA1 depletion on SSA. Data shown are the means of the four data points obtained from two independent experiments each performed in duplicates. Error bars represent SDs. ***$p<0.001$. (**B**) SSA activities of the BRCA1 variants relative to the wt protein. A schematic of the SA-GFP reporter assay is shown at the upper left corner. Data shown are the means ± SDs from two to five independent experiments for each variant or mutant. *$p<0.05$. See *Figure 2—source data 1* for details. (**C**) Effects of BRCA2 and PALB2 depletion on SSA. Data shown are the means ± standard errors of mean (SEMs) from four independent experiments. *$p<0.05$; **$p<0.01$. (**D**) Requirement of RAD52 for SSA upregulation following BRCA2 and PALB2 depletion. Data shown are the means ± SEMs from three independent experiments. *$p<0.05$; **$p<0.01$. (**E**) BRCA1 and BRCA2-binding defects of PALB2-L21A, L35A and A1025R mutants. The FLAG-HA-tagged mutants were transiently expressed in 293T cells and IPed with anti-FLAG M2 agarose beads. (**F–G**) HR (**F**) and SSA (**G**) activities of PALB2-L21A, L35A and A1025R mutants. Data shown are the means ± SDs from three to five independent experiments for each mutant. ***$p<0.001$. (**G**) Levels of PALB2 protein following knockdown

*Figure 2 continued on next page*

*Figure 2 continued*

and re-expression in U2OS/DR-GFP cells. Cells untreated with any siRNA were used as a control for the endogenous protein abundance.

The following source data is available for figure 2:

**Source data 1.** SSA activities of the BRCA1 variants and mutants analyzed in this study.

## Effects of BRCA1 missense variants on PARPi and cisplatin resistance

To profile the functional impact and therapeutic relevance of BRCA1 variants, we developed a relatively simple drug sensitivity assay (*Figure 3A–B*) using the BRCA1-deficient, triple negative MDA-MB-436 breast cancer cells (*Elstrodt et al., 2006*). Briefly, MDA-MB-436 cells were transfected with cDNA constructs expressing wt BRCA1 or the variants and then subjected to selection with either Geneticin (G418) alone (for transfection efficiency) or a combination of G418 and either olaparib, a potent PARP inhibitor (PARPi), or cisplatin. Colonies were allowed to grow for a period of 3–4 weeks. Under the condition used, we never obtained a single olaparib-resistant colony in untransfected or vector transfected cells; in only one (out of about ten) occasion, four cisplatin-resistant colonies were obtained from vector transfected cells. These results allow us to rule out any meaningful contribution of the endogenous truncated BRCA1 to our readout. Thus, MDA-MB-436 is a suitable cell line for BRCA1 functional analysis using our protocol.

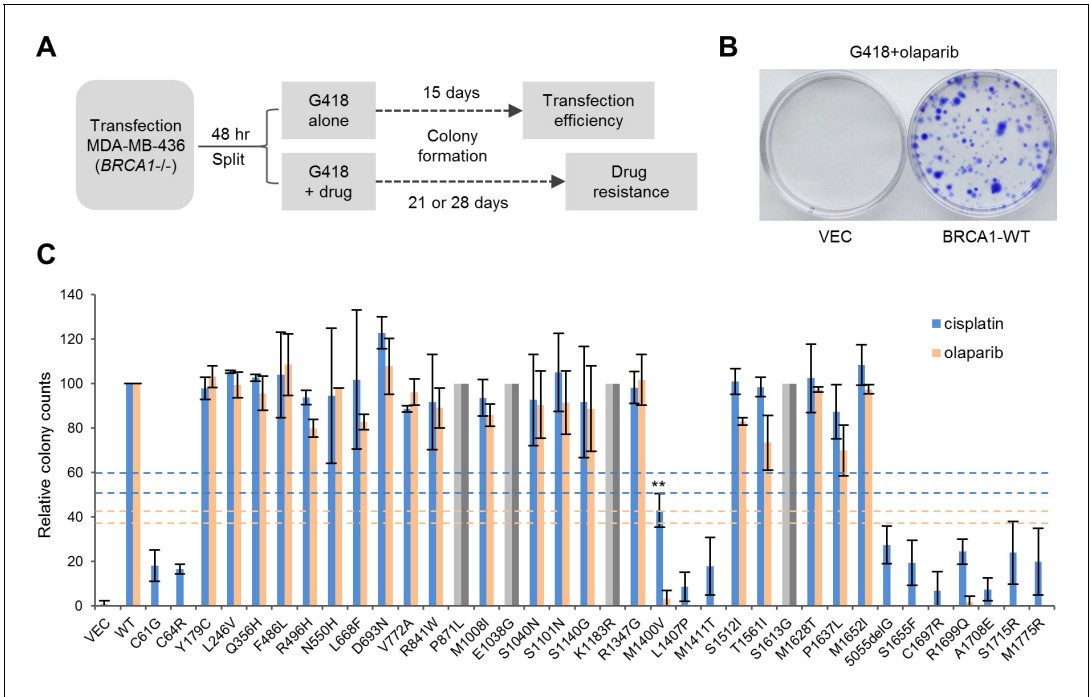

**Figure 3.** Abilities of the wt and mutant or variant BRCA1 proteins to confer cisplatin and olaparib resistance. (**A**) Schematic of the colony formation assay. The BRCA1 mutant MDA-MB-436 breast cancer cells were transfected with BRCA1 expression plasmids, reseeded and then selected with G418 alone or G418 with cisplatin or olaparib. Cells were stained with crystal violet 3–4 weeks after selection. (**B**) Representative crystal violet-stained plates. (**C**) Relative activities of wt and mutant or variant BRCA1 proteins to support colony formation in the presence of cisplatin or olaparib. Data shown are the means ± SDs from two to three independent experiments for each variant or mutant. **p<0.01. Horizontal lines represent the cutoff thresholds for cisplatin (blue) and olaparib (orange). See *Figure 3—source data 1* for details.

The following source data is available for figure 3:

**Source data 1.** Abilities of the BRCA1 variants and mutants to support olaparib and cisplatin resistance in MDA-MB-436 cells.

Resistance to PARPi was found to largely correlate with the HR activity of each variant (*Figure 3C*). Specifically, the BARD1-binding mutants (C61G and C64R), the PALB2-binding mutants (M1400V, M1407P and M1411T) and the BRCT mutants all demonstrated virtually no growth under PARP inhibition, whereas other VUSs were fully or largely resistant. Notably, BRCA1-M1400V, which showed only moderately impaired binding to PALB2 and 73% of the HR activity of the wt protein (*Figure 1B,D*), failed to confer PARPi resistance. The pattern of cisplatin resistance also largely reflected the HR activities of the variants, as the RING, CC and BRCT mutants all showed hypersensitivity, while other variants conferred greater than ~90% colony-forming ability of the wt protein. These data clearly demonstrate the correlation between HR activity, but not SSA activity, and drug resistance. They also underscore the importance of the RING, CC and BRCT domains and their corresponding binding partners for BRCA1 function in DNA repair.

## Localization defect of BRCT missense mutants

To understand the cause of the functional deficiency of the mutants more fully, we analyzed their subcellular localization in U2OS cells. Although the endogenous BRCA1 is almost exclusively localized in the nucleus, transiently expressed exogenous BRCA1 can be found in the cytoplasm in some cells, with the extent depending on the transfection condition. This is commonly explained by the insufficient amount of the endogenous BARD1, which is thought to shield the NES of BRCA1 thereby retaining BRCA1 in the nucleus (*Fabbro et al., 2002*). Indeed, BRCA1-C61G, which does not bind BARD1, showed a more diffuse localization pattern than the wt protein (*Figure 4A*). The PALB2-binding mutant M1411T behaved like the wt protein, as reported before (*Sy et al., 2009*). Interestingly, patient-derived mutations that truncate BRCA1 at the BRCT domain have been reported to cause BRCA1 to localize primarily in the cytoplasm (*Rodriguez et al., 2004*), even though the mutants retain intact NLSs and can still bind BARD1. The localization defect of the BRCT mutants cannot be rescued by Leptomycin B (LMB), which blocks CRM1-dependent export but can instead be rescued by co-expression of BARD1 (*Rodriguez et al., 2004*). In this vein, we tested the above aspects for the BRCT missense mutations and obtained the same results (*Figure 4B–E*). We consider the BRCT point mutants to be different from truncating mutants for two reasons: first, the majority of the these mutants were expressed at much lower levels than the wt protein presumably due to destabilization, whereas the truncating mutants were not (*Figure 1—figure supplement 2*), although frameshift mutations often cause nonsense-mediated decay (NMD) of mRNA in vivo; second, the missense mutants may still bind known interaction partners with low affinity or even certain unknown factors that may affect their localization. Collectively, the data lend strong support to the notion that the BRCT mutants may be unable to enter the nucleus and further suggest that the BRCT domain may bind a heretofore unidentified factor that promotes BRCA1 nuclear entry, with BARD1 directly or indirectly involved in the process.

To learn to what extent the lack of HR activity of the BRCT mutants is due to mislocalization, we tested their activities when co-expressed with exogenous BARD1. Co-expression of BARD1 slightly increased the HR activity of some of the mutants, but the overall effect was modest at best (*Figure 4F*). For example, mutant S1655F showed comparable, if not better, localization than the wt protein (*Figure 4E*), but still had a threefold lower HR activity. These data suggest that the lack of HDR activity of the BRCT mutants is due to both their localization defect and, perhaps more importantly, their inability to interact with one or more of the three binding partners, namely Abraxas, CtIP and BRIP1, or with another binding partner yet to be identified.

## Role of the BRCA1 I26A mutation and BRCA1 sumoylation in HDR and drug resistance

The BRCA1/BARD1 heterodimer exhibits robust ubiquitin E3 ligase activity in vitro (*Hashizume et al., 2001*; *Ruffner et al., 2001*). An artificial I26A mutation was isolated in a screen for residues in the RING domain that abrogate E3 ligase activity without affecting BARD1 binding (*Brzovic et al., 2003*). It was first thought that the E3 ligase activity may play an important role in DNA repair and/or tumor suppression; however, mouse cells carrying a I26A knockin allele did not show any significant defect in HR activity (*Reid et al., 2008*), and the mutant protein was still able to suppress mammary tumor development in *Brca1* conditional knockout models (*Shakya et al., 2011*). To our surprise, I26A substantially reduced, by about three fold, the coIP of endogenous BARD1

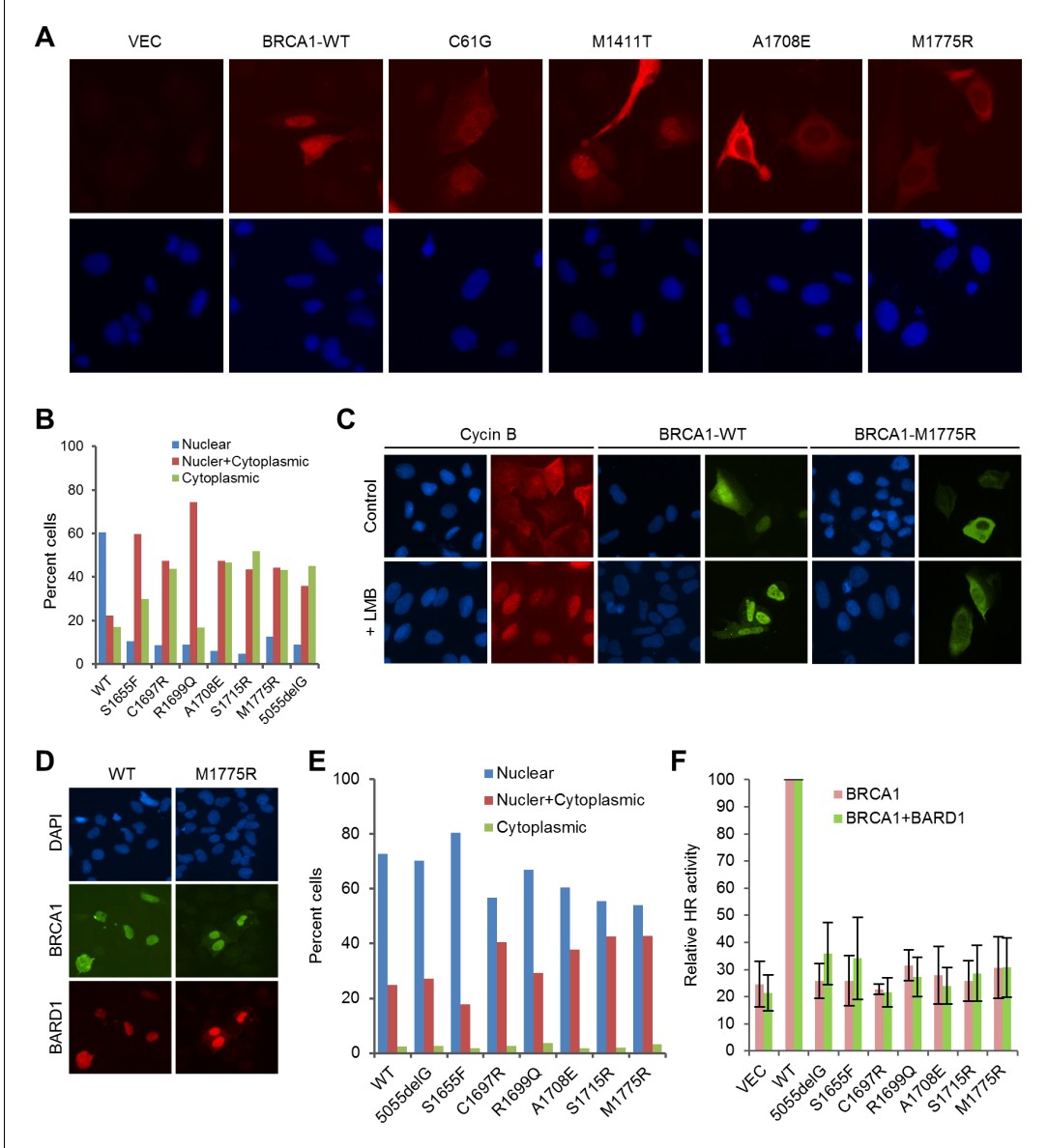

**Figure 4.** Rescue of the localization defect but not HR activity of BRCA1 BRCT missense mutants by BARD1. (A) Immunofluorescence staining of wt BRCA1 and representative RING, CC and BRCT mutants. 3xMyc-tagged BRCA1 expression constructs were transiently transfected into U2OS/DR-GFP cells first depleted of endogenous BRCA1, cells were fixed 48 hr after transfection, and the proteins were stained with an anti-Myc antibody. (B) Quantification of the subcellular distribution of wt BRCA1 and a panel of BRCT mutants. cDNA constructs were transfected into U2OS cells and the tagged BRCA1 proteins were stained as in (A). (C) Effect of Leptomycin B (LMB) on the localization of wt BRCA1 and BRCA1-M1775R. The BRCA1 proteins were expressed and stained as in (A). LMB (50 ng/ml) was added 48 hr after transfection, and cells were incubated with LMB for 12 hr prior to fixation. Staining of endogenous cyclin B in untransfected U2OS cells was used as a positive control. (D–E) Rescue of the localization defect of BRCA1 BRCT mutants by BARD1. 3xMyc-tagged BRCA1 and FLAG-HA-tagged BARD1 were transiently co-expressed in U2OS cells, cells were fixed 48 hr after transfection, and the proteins were stained with Myc and HA antibodies, respectively. Panel D shows the staining patterns of wt and two representative mutants of BRCA1 and the co-expressed BARD1. Panel E shows the quantification of the results. (F) No rescue of the HR defect of the BRCA1 BRCT mutants by BARD1 co-expression. U2OS/DR-GFP cells were depleted of the endogenous BRCA1 for 48 hr and then co-transfected with BRCA1 and BARD1 expression constructs. GFP-positive cells were scored another 52 hr later. The values were normalized against that of wt BRCA1, which was set as 1. Data shown are means ± SDs from three independent experiments.

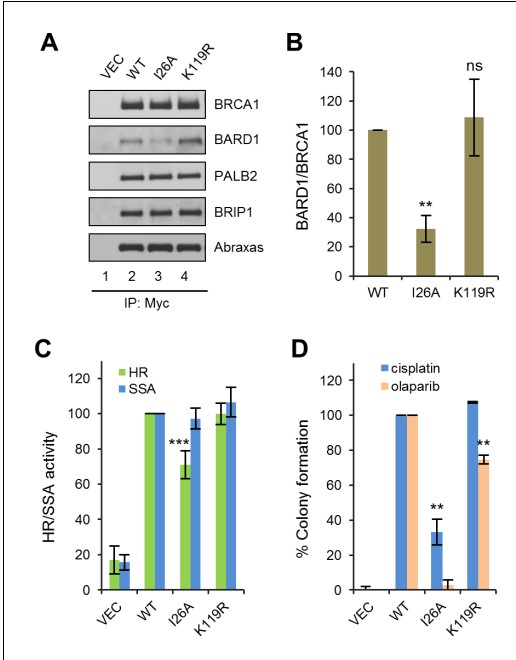

**Figure 5.** Effects of BRCA1 sumoylation and E3 ligase activity on HR and drug resistance. (**A**) Effect of I26A and K119R mutations on BRCA1 binding to BARD1 and other interacting partners. The proteins were transiently expressed in 293T cells and IPed with anti-Myc. (**B**) Quantification of the BARD1-binding capacity of the BRCA1 mutants. Data shown are the means ± SDs of the ratios of BARD1 and BRCA1 band intensities from four independent experiments. **p<0.01. (**C**) HR and SSA activities of the BRCA1 mutants relative to that of the wt protein. Data shown are the means ± SDs from two to six independent experiments for each mutant. ***p<0.001. See *Figure 1—source data 1* and *Figure 2—source data 1* for details. (**D**) Abilities of the BRCA1 mutants to confer resistance to cisplatin and olaparib. Data shown are the means ± SDs from three independent experiments for both mutants. **p<001. See *Figure 3—source data 1* for details.

with ectopically expressed BRCA1 (*Figure 5A*), indicating that this mutation significantly weakens the BRCA1-BARD1 interaction. Thus, I26A is a complex mutation affecting at least two different BRCA1 properties. Functionally, I26A reduced the HR activity of BRCA1 by ~30% (*Figure 5B*); although the apparent HR defect caused by I26A was mild, the mutation caused a 67% reduction in colony formation in the presence of cisplatin and virtually a complete loss of resistance to olaparib (*Figure 5C*).

It has been reported that BRCA1 can be sumoylated and this modification promotes the E3 ligase activity of the BRCA1/BARD1 complex (*Morris et al., 2009*). The same study identified lysine 119 (K119) as a critical residue for BRCA1-SUMO conjugation and the promotion of its E3 ligase activity. Thus, we generated and analyzed a K119R mutation in parallel with the I26A mutant. K119R had no effect on BRCA1 HR, nor did it affect cisplatin resistance (*Figure 5B–C*); however, colony formation in the presence of olaparib was reduced by ~30%, suggesting a possible HR-independent role of BRCA1 sumoylation for PARPi resistance.

## Deletion analyses of BRCA1 for HDR and drug resistance

To systematically assess the contribution of various BRCA1 structural elements in HDR and drug resistance, we generated a series of 10 overlapping deletions (*Figure 6A*). Interestingly, BD1, which lacks both the RING domain and the NES, was expressed at a much higher level, and despite the complete lack of the RING domain, it still associated with a small but significant amount of BARD1 (*Figure 6B*); it showed ~50% activity for both HR and SSA (*Figure 6C*), which is significantly higher than the activities of either C61G or C64R mutants (*Figure 1D*). As expected, BD8, which lacks the CC motif, was unable to bind PALB2, and BD10 lacking the BRCT domain failed

to bind Abraxas and BRIP1. In line with observations made with the point mutants (*Figures 1D* and *2A*), BD8 showed no HR activity and a threefold increase in SSA, while BD10 displayed similar HR and SSA activities as the vector. Notably, BD4, lacking both NLSs, still retained ~75% activity in both HR and SSA assays, indicating that a significant amount of BRCA1 can still be recruited into the nucleus. In contrast, BD7, which retains all known functional domains, showed only ~50% of HR and SSA activities, suggestive of the possible existence of a novel element in the deleted region that is important for DNA repair.

Consistent with their lack of HR activity, BD8 and BD10 were completely or nearly completely unable to confer resistance to either drug (*Figure 6D*). Although BD1, BD4 and BD7 all displayed 50–75% apparent HR activity in the reporter assay, they supported no more than 20% colony formation. Also, BD9 only supported ~50% of colony formation despite having no defect in the aforementioned protein interactions and possessing greater than 80% apparent HR and SSA activities. These observations demonstrate a positive but non-linear correlation between apparent HR activity and drug resistance. The non-linearity could be due to different effects of the deletions on protein

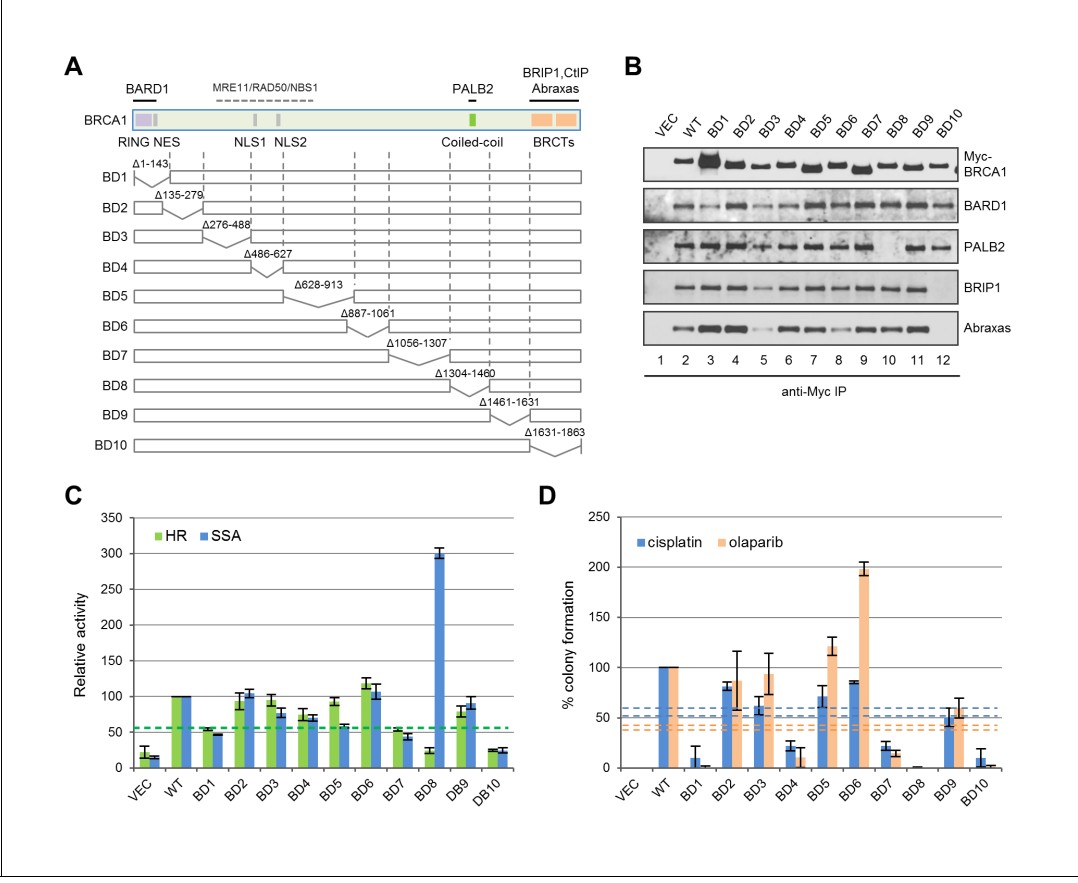

**Figure 6.** Functionalities of BRCA1 deletion mutants in HR, SSA and drug resistance. (**A**) Schematic of wt BRCA1 and 10 overlapping deletions generated for this study. (**B**) Capacity of the deletion mutants in binding key interacting partners. The proteins were transiently expressed in 293T cells and IPed with anti-Myc. (**C**) HR and SSA activities of the deletion mutants. See *Figure 1—source data 1* and *Figure 2—source data 1* for details. (**D**) Levels of cisplatin and olaparib resistance conferred by the deletion mutants. Values presented are means ± SDs from two to four independent experiments for each mutant. See *Figure 3—source data 1* for details.

folding, stability, conformation, posttranslational modifications and dynamics in protein-protein interactions; it may also reflect the differences between the two cell lines (U2OS/DR-GFP vs MDA-MB-436), time frames used or the different nature of I-SceI and drug-induced DNA breaks. Surprisingly, BD6 was twice as effective as the wt protein in supporting colony formation in the presence of olaparib but less effective in the presence of cisplatin, indicating a significant difference in the mechanisms of resistance to the two drugs despite a common requirement for HR.

## Discussion

In this study, we systematically assessed HR and SSA activities of 32 natural variants, including 11-patient-derived missense mutations that affect known protein-protein interactions and the top 22-patient-associated missense variants, as well as 10 overlapping deletion mutations that span the entire length of the protein. Our data clearly demonstrate that the HR function of BRCA1 requires the RING, CC and BRCT domains. Interestingly, deletion of residues 1056–1357 (BD7) caused ~50% reduction in both HR and SSA activity without affecting the binding of any of the above known partners. This segment of BRCA1 does not contain any recognizable domains and has not been implicated in HR before, suggesting the existence of a novel binding partner or regulatory mechanism. Together, these results establish a functional landscape of BRCA1 for its HDR function.

The RING domain of BRCA1 binds to the RING domain of BARD1, and the BRCA1/BARD1 heterodimer formation is critical for their stability, their E3 ubiquitin ligase activity and the nuclear

retention of BRCA1 (*Fabbro et al., 2002*; *Hashizume et al., 2001*; *Ruffner et al., 2001*). Conditional knock out of either *Brca1* or *Bard1* in murine mammary epithelial cells led to the development of mammary carcinomas that are indistinguishable from each other (*Shakya et al., 2008*), and the BRCA1 C61G mutant that abrogates BARD1 binding failed to suppress mammary tumor development in mice (*Drost et al., 2011*). Moreover, the I26A mutation that abolishes its E3 ligase activity did not affect HR activity or mitomycin C (MMC) sensitivity (*Reid et al., 2008*), nor did it affect tumor suppression in mice (*Shakya et al., 2011*). These findings suggested that the BARD1-binding function of the RING domain, rather than its E3 ligase activity, plays a key role in BRCA1 HR and tumor suppression activity. In this regard, our findings on the I26A mutation are worth noting. First, it caused a significant impairment in BARD1 binding (*Figure 5A*); second, it moderately reduced HR efficiency (*Figure 5B*); and third, it caused sensitivity to both cisplatin and olaparib under the condition used (*Figure 5C*). The reduced BARD1 binding is consistent with the fact that I26A mutant mouse cells had substantially reduced amount of both BRCA1 and BARD1 (*Reid et al., 2008*). Also, a recent report showed that HeLa cells expressing only BRCA1-I26A were similarly sensitive to olaparib as BRCA1 knockdown cells (*Densham et al., 2016*). Thus, it is safe to conclude that the I26A mutation affects HR and drug resistance in human cells. The discrepancy in observed HR activities and drug sensitivities in mouse ES cells and human cancer cells may be explained by the different tolerance for partial loss of BARD1 binding in the two systems or the presence of other mutations in cancer cells. At the same time, although I26A mutant mouse cells showed no defect in apparent HR activity or MMC resistance, reduced gene targeting efficiency and increased chromosomal abnormalities after MMC treatment were noted (*Reid et al., 2008*), indicative of a possible defect in HR in mouse cells as well. Despite the discrepancy, it should be noted that our results do not contradict the argument that the E3 ligase activity of BRCA1 is dispensable for tumor suppression (*Shakya et al., 2011*).

Mutations of the BRCT domain dramatically abolish HR activity, and the mechanism is also complex. First, this domain directly interacts with at least three proteins, Abraxas, a scaffold protein (*Wang et al., 2007*), BRIP1, a DNA helicase (*Cantor et al., 2001*), and CtIP, an endonuclease involved in resection (*Sartori et al., 2007*). It has recently been shown that CtIP binding to BRCA1 is dispensable for CtIP-mediated DNA resection (*Polato et al., 2014*; *Reczek et al., 2013*), yet to what extent BRCA1 HR activity depends on Abraxas and BRIP1 remains unclear. Second, mutation of the BRCT domain causes a major defect in the nuclear accumulation of BRCA1 (*Figure 4A–B*), supporting a key role of the BRCT domain for BRCA1 nuclear entry. Our finding that BARD1 can significantly rescue nuclear localization of BRCT mutants but not their HR defects (*Figure 4F*) also supports an important and direct role of the BRCT domain for BRCA1 DNA repair activity. Collectively, our data demonstrate the dual role of the BRCT domain and underscore a complex regulation of BRCA1 localization and repair function by the RING and BRCT domains and their binding partners (*Figure 7A*).

Consistent with early observations made in mouse cells (*Stark et al., 2004*), BRCA1 depletion reduced the efficiency of both HR and SSA, whereas depletion of BRCA2 led to severe loss of HR (data not shown) but increased SSA (*Figure 2B*). As efficient resection is necessary for both repair mechanisms, the data is consistent with the notion that BRCA1 plays a key role in resection (*Bunting et al., 2010*). Importantly, we found that depletion of PALB2 showed similar, if not more dramatic, induction of SSA as did BRCA2 depletion (*Figure 2B*). Moreover, all three point mutations in BRCA1 that affect PALB2 binding led to increased SSA usage (*Figure 2A*), and two different PALB2 mutants with greatly impaired BRCA1-binding capacity both failed to suppress the upregulation of SSA in PALB2-depleted cells (*Figure 2F*). These results clearly demonstrate that the direct interaction between BRCA1 and PALB2 is required for suppressing, or at least preventing, SSA, while promoting HR (*Figure 7A–B*). As such, our results support two critical roles of BRCA1 function in DSB repair, promoting resection and recruiting/stimulating PALB2, which then channels the ssDNA down the HR path (*Figure 7B*). Based on the available data, PALB2 is the key partner of BRCA1 specifically for its HR function and the primary switch point, acting upstream of BRCA2, for HDR pathways following resection.

VUSs are commonly found during clinical genetics tests; however, their biological and clinical significance is often difficult to ascribe with confidence, and this uncertainty poses significant challenges for both clinicians and patients. Our results demonstrate that relatively common VUSs outside the RING and BRCT domains do not significantly affect the HDR activity of BRCA1. Our findings are

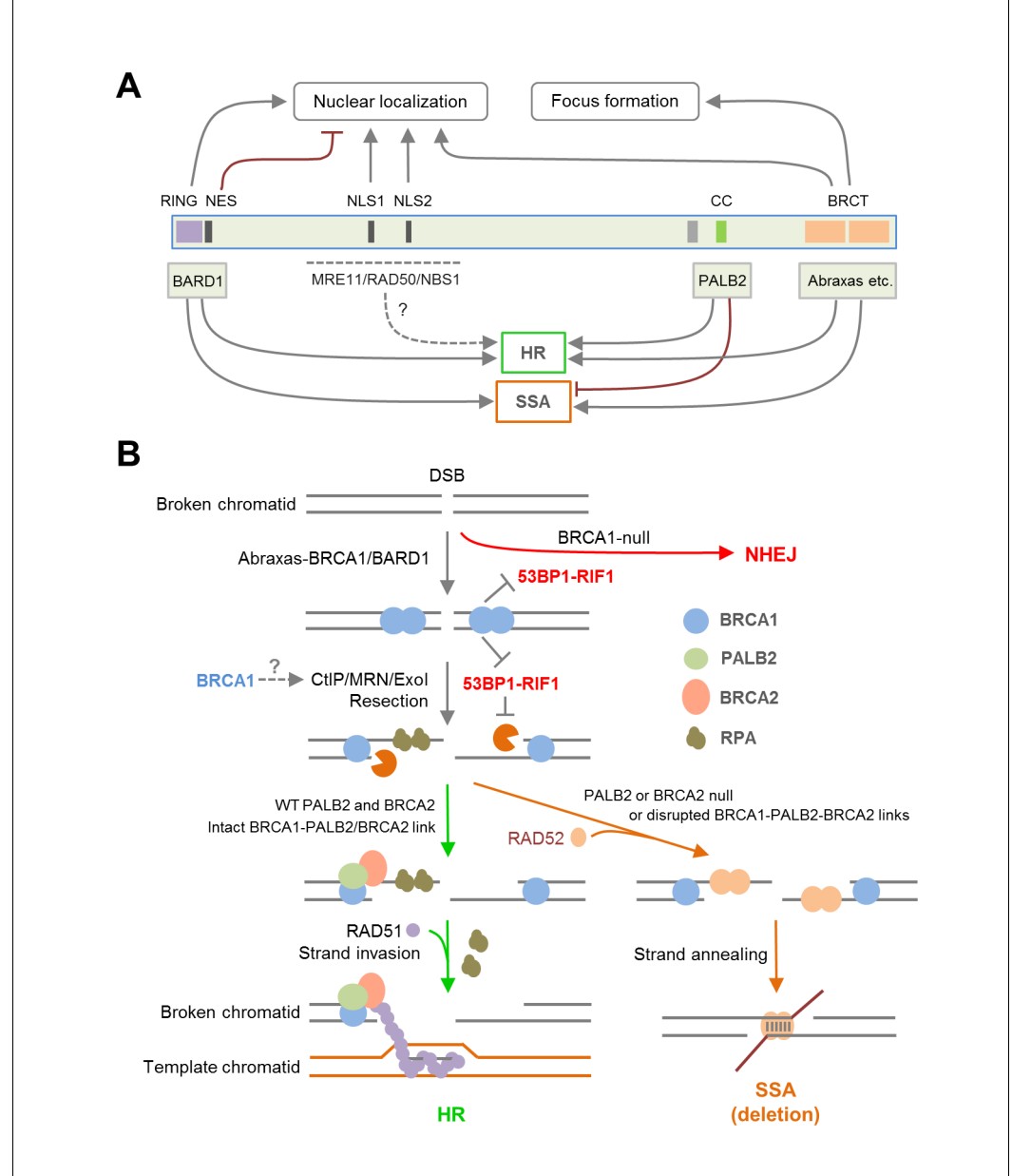

**Figure 7.** Summary models of the regulation and function of BRCA1 in HR and SSA. (**A**) Roles of BRCA1 structural elements and binding partners on its nuclear localization and HDR (HR and SSA) activities. In brief, the NLSs and BRCT domain both promote BRCA1 nuclear entry, whereas the NES mediates BRCA1 export from the nucleus. BARD1 bound to the RING domain shields the NES of BRCA1 thereby promoting its nuclear retention. The RING and BRCT domains are required for both HR and SSA, whereas the CC domain promotes HR but inhibits or prevents SSA through its binding to PALB2. (**B**) A model of the BRCA1-PALB2-BRCA2 axis in the regulation of HR and SSA. Following DSB formation, the BRCA1/BARD1 complex was recruited to DNA damage sites by Abraxas and perhaps another yet to be defined factor(s). The presence of BRCA1 at damage sites promotes resection and inhibits non-homologous end joining (NHEJ), at least in part, by counteracting the resection-inhibiting activity of the 53BP1-RIF1 complex. At the same time, BRCA1 helps recruit the PALB2/BRCA2 complex, which displaces RPA from the resected ssDNA ends and loads RAD51 to initiate HR. When PALB2 or BRCA2 is lost or when the direct interactions in the BRCA1-PALB2-BRCA2 axis are disrupted, RAD52 binds to resected ends and mediates SSA when homologous sequences are available, leading to genomic deletions. It is unclear whether PALB2 or BRCA2 can directly suppress the binding of RAD52 to ssDNA or inhibiting its strand annealing activity.

mostly consistent with several previous studies (Table 2), including the above noted mouse ES cell-based study that covered 6 of the top 22 variants (Bouwman et al., 2013); on the rare PALB2-binding mutants L1407P and M1411T, however, our results showed that they are severely defective in HR (Figure 1D) and conferring drug resistance (Figure 3C), whereas their effects in mouse cells were less pronounced. Moreover, M1400V, which was found to be neutral in the mouse cell system, shows clear sensitivity to olaparib treatment in the human cell line. This discrepancy could again be due to a difference in the thresholds of human and mouse cells to tolerate partial loss of BRCA1-PALB2 interaction or the presence of other mutations in the human cancer cells.

Platinum-based therapeutics have been commonly used for treatment of ovarian cancer, and olaparib has recently been approved for the treatment of *BRCA1/2* mutant ovarian cancer, with multiple trials on breast and other cancers ongoing or recently completed. These agents generate lesions that block or collapse DNA replication forks, leading to DSBs that require HR and therefore BRCA1/2 for repair (Lord and Ashworth, 2016). Under the condition used, olaparib appeared to have a better selectivity in killing MDA-MB-436 cells expressing mutant BRCA1 proteins. Although this could be due to a slightly lower than optimal concentration of cisplatin used, the possibility that cisplatin is intrinsically less selective than olaparib cannot be ruled out. Based on data from this study and other recent studies, patients with relatively common missense mutations in the RING and BRCT domains, such as C61G and A1708E, respectively, or rare mutations in the CC motif, such as L1407P or M1411T, are likely to respond to platinum- or PARPi-based therapies, unless resistance mechanisms have already occurred prior to treatment or are induced by the treatment (Lord and Ashworth, 2013). However, except for C61G and A1708E, all other top 22 missense variants are fully or mostly functional both in HR and in conferring cisplatin and olaparib resistance (Figure 3C), which corroborates with clinical genetics evidence to further cast doubts on their pathogenicity and diminish the likelihood of a positive response of the patients to the above regimens. Based on the results of our deletion analyses and other available data, it is unlikely for any VUS outside the RING, CC and BRCT domains to be strongly pathogenic or to elicit high sensitivity to platinum or PARPi therapies (unless it destabilizes the transcript or protein). However, the region deleted in BD7 (residues 1056–1,307) may harbor VUS that have intermediate effects.

## Materials and methods

### Cell culture

U2OS/DR-GFP HR reporter cells were described before (Nakanishi et al., 2005; Xia et al., 2006). U2OS/SA-GFP SSA reporter cells (Gunn and Stark, 2012) were a gift from Dr. Jeremy Stark. U2OS and 293T cell lines were purchased from American Type Culture Collection (ATCC). These four cell lines were cultured in Dulbecco's Modified Eagle's Medium (DMEM) supplemented with 10% fetal bovine serum (FBS) and 1x Penicillin-Streptomycin. MDA-MD-436 cells was cultured in DMEM/F12 (1:1) supplemented with 10% FBS and 1x Penicillin-Streptomycin. All cells were cultured in a humidified chamber with 5% $CO_2$ at 37°C. Mycoplasma was not tested, but the cells have been cultured in the presence of Plasmocin (ant-mpt, InvivoGen) to eliminate potential mycoplasma contamination. No commonly misidentified cell lines were used.

### Plasmids and mutagenesis

All BRCA1 expression constructs were based on pcDNA-3xMyc-BRCA1 (Chen et al., 1998). The pOZ-FH-C1-PALB2 vector expressing FLAG-HA-tagged PALB2 was described before (Ma et al., 2012). The pOZ-FH-N-BARD1 expressing FLAG-HA-tagged BARD1 was described before (Greenberg et al., 2006). All mutations or deletions were generated through site-directed mutagenesis following the QuikChange protocol (Agilent Technologies).

### cDNA transfection, immunoprecipitation (IP) and western blotting

For testing protein-protein interaction, BRCA1 expression constructs were transfected into 293T cells plated at a density of $5 \times 10^5$ cells per well in six-well plates and transfected with 2 µg of plasmid per well using FuGENE HD or X-tremeGENE 9 XP. Cells were collected 30 hr after transfection and lysed with 350 µl of NETNG-300 (300 mM NaCl, 1 mM EDTA, 20 mM Tris-HCl, 0.5% Nonidet P-40% and 10% Glycerol) containing Complete protease inhibitor cocktail (Roche). The 3xMyc-

tagged BRCA1 proteins were IPed for 3–4 hr with anti-Myc (9E10, Covance) and protein A-agarose beads (Roche). For western blotting analyses, proteins were resolved on 4–12% Tris-Glycine SDS gels, transferred onto nitrocellulose membranes and probed with the following antibodies- Myc (9E10, Covance), PALB2 (M11) (*Xia et al., 2006*), BARD1 (H300, Santa Cruz), BRIP1 (a gift from Dr. Sharon Cantor, University of Massachusetts Medical School), Abraxas (ab139191, AbCam). The secondary antibodies used were Horseradish peroxidase (HRP)-conjugated sheep anti-mouse IgG (NA931V, GE Healthcare) and donkey anti-rabbit IgG (NA9340V, GE Healthcare). Immobilon Western Chemiluminescent HRP Substrate (Millipore) was used to develop the blots.

## Measurement of HR and SSA efficiency

To measure the HR and SSA activities of the BRCA1 and PALB2 variants, U2OS/DR-GFP and U2OS/SA-GFP cells were plated in 10 cm dish at $1.2 \times 10^6$ cells per dish and allowed to adapt overnight. Cells were transfected with an siRNA targeting BRCA1 3'-UTR using Lipofectamine RNAiMax (Life Technologies) (10 nM final concentration of siRNA and 12 μl of RNAiMax per plate). Cells were split into six-well plates (200,000 cells per well) 30 hr after transfection. After another 18 hr, cells were co-transfected with BRCA1 or PALB2 expression constructs (1 μg) and the I-SceI expression plasmid pCBASce (1.5 μg) using 6 μl of X-tremeGene 9 (Roche). Cells were harvested 54 hr post the second transfection, and GFP-positive cells were counted by fluorescence-assisted cell sorting (FACS). The sense strand sequence of the siRNA is GGAUCGAUUAUGUGACUUAdTdT.

To measure SSA activity after BRCA2, PALB2 or RAD52 knockdown with different siRNAs, U2OS/SA-GFP cells were seeded into six-well plates at density of 175,000 cells per well and transfected with siRNAs ~18 hr later using using Lipofectamine RNAiMax (Life Technologies) (10 nM final concentration of each siRNA and 3 μl of RNAiMax per well). Media were refreshed 24 hr post transfection, and cells were transfected again with pCBAcse with X-tremeGene 9 another 24 hr later (48 hr after the first transfection). Cells were collected and subjected to FACS analysis ~52 hr after the second transfection. See *Supplementary file 1* for the sequences of the siRNAs used.

## Immunofluorescence

Cells were seeded onto coverslips in 12-well plates the day before transfection. Cells were transfected with 1 μg BRCA1 expression constructs using 2.5 μl of FuGENE 6 or X-tremeGENE 9 (Roche). At 48 hr after transfection, cells were fixed with 3% (w/v) paraformaldehyde (in PBS with 300 mM sucrose) for 10 min at room temperature, permeabilized with 0.5% Triton X-100 (in PBS) and then sequentially incubated with primary and secondary antibodies (diluted in PBS containing 5% goat serum) for 1 hr each at 37°C. Each of the above steps was followed by three PBS washes. After staining, coverslips were mounted onto glass slides with VECTASHIELD Mounting Medium with DAPI (Vector Labs) and observed using a fluorescent microscope. The primary antibodies used were anti-Myc (9E10, Covance) and anti-BRCA1 (#07–434, Millipore).

## Assay for PARPi and cisplatin resistance

MDA-MB-436 cells were seeded into six-well plates at a density of $1 \times 10^6$ cells per well the day before transfection. Cells were transfected with 1 μg of empty vector or 2 μg of BRCA1 expression vectors using 6 μl of X-tremeGENE 9 (Roche). Cells were trypsinized 36 hr after transfection and reseeded into 10 cm plates at 100,000 cells per plate in a volume of 10 ml. Another 16 hr later, 1 ml of 10X G418 (6 mg/ml dissolved in the same culture medium) or 10X G418 containing 2.2 μM cisplatin or olaparib was added to each plate. The final concentrations of G418, cisplatin and olaparib were 550 μg/ml, 200 nM and 200 nM, respectively. All experiments were performed in duplicate plates for each construct. For cells selected with G418 alone, 15 days after selection, one plate was trypsinized and cells counted with a Vi-CELL Cell Counter (Beckman Coulter) to determine viable cell number, and the other plate was stained with Crystal Violet (0.5% w/v in 95% ethanol, 5% acetic acid) to count the number of colonies. For cells selected with both G418 and cisplatin, both plates were stained 21 days after selection. For cells subjected to G418 and olaparib double selection, both plates were stained 28 days after selection. Colonies with approximately 50 cells or more were counted, and the numbers were normalized against the number of viable cells on corresponding plates with G418 alone. The vast majority of constructs were measured by three or more independent experiments (each with duplicate sets of plates), with the only exceptions being when the first

two experiments produced nearly identical results or were carried out using two independent plasmids.

## Sensitivity and specificity analyses

We assessed the sensitivity and specificity of our assays using the Receiver Operating Characteristics (ROC) analysis to simultaneously capture the maximum sensitivity and specificity of the assays with 95% confidence interval as previously described (*Guidugli et al., 2013*). The sensitivity of each assay was defined as the ability to detect variants retaining wt activity of BRCA1 in the respective assays and specificity the ability to detect variants showing loss of BRCA1 function. For the calculation of specificity and sensitivity of our assays, we compared the values of benign variants (Align-GVD grade of C0 and IARC classification of 1) and pathogenic variants (Align-GVD grade of C35-C65 and IARC classification of 5) that have been confirmed by previous functional studies (*Tables 1* and *2*). We also included values of the wt cDNA in the benign group and the vector as well as a bona fide pathogenic mutation 5055delG (p.Val1646Serfs or L1657STOP) in the pathogenic group. Threshold of each assays were defined individually. To rule out variability in the assays, we also calculated the sensitivity and specificity using the highest observed values for the pathogenic variants and the lowest observed values for the benign variants with the cut off used to define either the upper or lower limit of the threshold. For each of the assays, dotted lines were drawn to represent the cut off threshold that captures the specificity and sensitivity of the assays at 100%, respectively. Analysis was not performed for the SSA assay as elevated SSA activity is associated with PALB2-binding mutants, which has yet to be defined, while loss of SSA is associated with BRCT and RING domain mutations. All analyses were conducted using GraphPad Prism 5.0 (GraphPad Software).

## Statistical analysis

Raw data were normalized against values of wt BRCA1, wt PALB2, control siRNA (NSC1) or the empty vector, where applicable. Statistical significance was calculated using Students' *t* test. p-Values smaller than 0.05 were considered significant.

## Acknowledgements

We thank Drs. Jeremy Stark (City of Hope) for providing the U2OS/SA-GFP reporter cells, Sharon Cantor (Univ. of Massachusetts) for the BRIP1 antibody and Roger Greenberg (Univ. of Pennsylvania) for the pOZ-FH-N-BARD1 plasmid. We also thank Dr. Jean-Yves Masson (Laval University, Quebec) for sharing unpublished results. This work was supported by the National Cancer Institute (R01CA138804 and R01CA188096 to BX, R01CA169182 to SG and R01CA195612 to ZS) and by the Flow Cytometry Core Facility of Rutgers Robert Wood Johnson Medical School, a Shared Resource of Rutgers Cancer Institute of New Jersey (P30CA072720) and NIH Shared Instrumentation Grant (1 S10 RR025468). RWA was a recipient of a postdoctoral fellowship from the Department of Defense Breast Cancer Research Program (W81XWH-10-1-0486).

## Additional information

### Funding

| Funder | Grant reference number | Author |
|---|---|---|
| National Cancer Institute | R01CA138804 | Bing Xia |
| Congressionally Directed Medical Research Programs | W81XWH-10-1-0486 | Rachel W Anantha |
| National Cancer Institute | R01CA188096 | Bing Xia |
| National Cancer Institute | R01CA169182 | Shridar Ganesan |
| National Cancer Institute | R01CA195612 | Zhiyuan Shen |

The funders had no role in study design, data collection and interpretation, or the decision to submit the work for publication.

## Author contributions
RWA, Data curation, Formal analysis, Investigation, Methodology, Writing—original draft, Writing—review and editing; SS, TKF, Data curation, Formal analysis, Investigation, Methodology, Writing—review and editing; SM, Data curation, Investigation, Writing—review and editing; JL, Resources, Investigation; ZS, Resources, Funding acquisition, Writing—review and editing; SG, Supervision, Funding acquisition, Writing—review and editing; BX, Conceptualization, Resources, Data curation, Formal analysis, Supervision, Funding acquisition, Validation, Investigation, Methodology, Writing—original draft, Project administration, Writing—review and editing

## Author ORCIDs
Tzeh Keong Foo, http://orcid.org/0000-0003-0168-7054
Zhiyuan Shen, http://orcid.org/0000-0003-2834-0309
Bing Xia, http://orcid.org/0000-0003-3259-6139

## Additional files

### Supplementary files
• Supplementary file 1. Sequences of siRNAs used in this study.

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
