## [Decision Letter]

Thank you for submitting your article "Functional and mutational landscapes of BRCA1 for homology-directed repair and therapy resistance" for consideration by *eLife*. Your article has been reviewed by three peer reviewers, and the evaluation has been overseen by Charles Sawyers as the Senior Editor and Reviewing Editor. The reviewers have opted to remain anonymous.

The reviewers have discussed the reviews with one another and the Reviewing Editor has drafted this decision. We are willing to consider a revised submission but only if it addresses the major concerns listed below under Essential Revisions.

Summary:

The authors have performed a comprehensive analysis of BRCA1 patient-derived mutations (VUS) for their effects on two DNA double-strand break (DSB) repair pathways, homologous recombination (HR) and single-strand annealing (SSA), and sensitivity to cisplatin and the PARP inhibitor olaparib. Mutations were examined along the length of the protein, including those falling in three established domains, the RING which interacts with partner protein BARD1, the coiled-coil domain which interacts with PALB2, and the BRCT which interacts with phosphoproteins, including Abraxis and BRIP1. The authors confirm that interactions with partner proteins are disrupted with mutations in these various domains. A total of 32 mutations were analyzed, including 11 with known functional consequences.

The authors make good use of established GFP reporters for HR and SSA. The duality of these reporters is notable in that proteins like BRCA1 that impair the first step of HR and SSA, end resection, are defective in both DSB repair pathways, while those like BRCA2 that impair downstream steps are decreased for HR and increased for SSA (Stark et al. 2004). The authors demonstrate that mutations that abrogate binding to key interacting proteins reduce HR and increase sensitivity to olaparib and cisplatin. Similarly, mutations that disrupt the RING or BRCT show the typical BRCA1 phenotype of a concomitant reduction in SSA, but those that disrupt PALB2 binding show a typical BRCA2-like phenotype of increased SSA (as does loss of PALB2 or binding to BRCA1). Thus, one could infer from these results that the first group is defective in end resection, while the second group is proficient at end resection but deficient in downstream steps (expected to be recruitment of BRCA2 based on previous reports). Deletion analyses largely support these conclusions, although there were some unexplained differences.

Overall this manuscript provides a comprehensive analysis of BRCA1 functional domains and patient-derived missense mutations for their role in DSB repair and response to chemotherapeutics, and differentiates the role of BRCA1 binding of PALB2 from that of other interacting proteins in promoting HR, while suppressing mutagenic SSA.

Essential revisions:

1) Novelty concerns and use of screening assays with appropriate citations

This manuscript presents a comprehensive mutation analysis, and so it is good to include mutations that have been analyzed previously. However, it is difficult to parse what information is new from that which is confirmatory of previous reports. The authors should therefore indicate which VUS have been analyzed before in functional assays.

The HR/SSA assay system is not entirely novel. Previously, Ransburgh et al. (Cancer Research 2010) have already measured the effects of BRCA1 sequence variants on HR in human cancer cells using siRNAi of endogenous BRCA1 and complementation by transiently expressed cDNA. Although the integration of the SSA reporter assay is novel, Ransburgh et al. should be cited to acknowledge the fact that they developed and used a similar assay system.

Increased SSA (and MMEJ) has recently been shown in lymphoblastoid cells from heterozygous PALB2 mutation carriers (Obermeier et al., Oncogene 2016).

Also the analysis of the effects of a series of BRCA1 deletion mutants on HR, SSA and the response to olaparib or cisplatin is an extension of previous work (Ransburgh et al., Cancer Research 2010).

The data in Figure 4 confirm previous findings by Rodriguez et al. (Exp Cell Res 2004) and nicely show that the defect of BRCT mutants in HR is not primarily caused by mislocalization of overexpressed mutant BRCA1. It is interesting to speculate why BRCT mutants do not seem to enter the nucleus efficiently. However, as this has been published before, Figure 4 could be moved to supplemental data.

2) Specificity and sensitivity of the assays

Functional analysis of BRCA1 VUS is important to aid clinical decision-making. RNAi of endogenous BRCA1 and transient overexpression of cDNA constructs may not capture all aspects of BRCA1 biology, but when the specificity and sensitivity of the assays are high, the functional data are relevant for genetic counseling. In addition to the controls mentioned by the authors, there are a number of variants known to be pathogenic or neutral (the BIC database needs to be updated). The ENIGMA consortium has published lists of these variants in Millot et al. (Hum Mut 2012). The authors should indicate all known controls and use these to calculate the specificity and sensitivity of their assays.

3) Truncated BRCA1 in MDA-MB 436 cells

The authors used MDA-MB 436 cells to analyze the abilities of BRCA1 variants to confer cisplatin or olaparib resistance. It should be noted that MDA-MB 436 cells have a BRCT truncating mutation and express a near full-length BRCA1 protein that can be stabilized and confer cisplatin and PARPi resistance (Johnson et al., PNAS 2013). It is therefore not a BRCA1-null cell line and results have to be evaluated with caution. The authors should include a paragraph on the limitations of their assays systems in the discussion, and cite the PNAS paper from Johnson et al.

4) BRCA1 and PALPB2 protein expression levels

The assays rely on artificial overexpression of BRCA1 variants, which may affect the results. How does the expressed BRCA1 protein level compare with the endogenous level? Preferably, BRCA1 needs to be expressed at equal and physiologically relevant levels in the relevant cell lines and assays. If it is heavily overexpressed it could be swamping the system, e.g. diluting interacting proteins, which does become relevant later in the paper. To address this issue, the authors should show expression levels of endogenous and overexpressed BRCA1 proteins.

How does the expressed PALB2 protein level compare with the endogenous level (Figure 2)?

5) Variable knockdown efficiencies and potential off-target effects of siRNAs

For the assays in U2OS cells, it is essential that there is efficient knockdown of endogenous BRCA1 in all cases. Knockdown efficiency should therefore be shown.

Figure 2 shows that knockdown of BRCA2 with 2 different siRNAs gives very different quantitative results: a 30-fold increase in SSA for siRNA 1949 and an 8-fold increase for the 9025. Even the PALB2 siRNAs give very high SSA increases. RAD51 protein levels are frequently affected by siRNAs (as shown by Elledge and colleagues), so it is important to show that RAD51 levels are not affected by the various siRNAs, especially 1949, which appears to be especially effective at increasing SSA while not being demonstrably better at knocking down BRCA2. Related to this, the less effective siRNA for PALB2 seems to be more effective at increasing SSA.

Although PALB2 knockdown results in more than 10-fold increase in SSA activity (Figure 2), re-expression of PALB2 only suppresses SSA less than 50% (Figure 2). What is the reason for this discrepancy?

6) Statistical analysis

At present, it is unclear which of the observed differences are significant. Statistical analysis is lacking as are source data files and both should be submitted.

---

## [Author Response]

*Essential revisions:*

*1) Novelty concerns and use of screening assays with appropriate citations*

*This manuscript presents a comprehensive mutation analysis, and so it is good to include mutations that have been analyzed previously. However, it is difficult to parse what information is new from that which is confirmatory of previous reports. The authors should therefore indicate which VUS have been analyzed before in functional assays.*

We acknowledge the reviewers’ concerns. We have added a Table 2 to include functional results of previous studies on any of the VUSs that are analyzed in our study with respect to HR activity and drug sensitivity. Table 1 has also been updated incorporating information from ClinVar and other databases.

*The HR/SSA assay system is not entirely novel. Previously, Ransburgh et al. (Cancer Research 2010) have already measured the effects of BRCA1 sequence variants on HR in human cancer cells using siRNAi of endogenous BRCA1 and complementation by transiently expressed cDNA. Although the integration of the SSA reporter assay is novel, Ransburgh et al. should be cited to acknowledge the fact that they developed and used a similar assay system.*

We apologize for not including Ransburgh et al. in our citation but we did cite Towler et al. (2013) that utilized a HR/SSA system developed by the same group. We have now added a citation for Ransburgh et al. as well. As far as we know, we might have been the first in the BRCA field to employ such transient protein replacement strategy to measure HR activity (Xia, et al., Mol Cell, 2006), although it was used to measure PALB2 activity, and Junjie Chen’s group appears to be the first to use the strategy for BRCA1 variants (Sy et al., PNAS, 2009). Also, we now use a newly designed siRNA targeting the 3’-UTR of BRCA1, and the protocol used in the current study is also different. Still, to avoid potential and unnecessary conflicts, we have changed the wording in the abstract from “developed” to “employed”.

*Increased SSA (and MMEJ) has recently been shown in lymphoblastoid cells from heterozygous PALB2 mutation carriers (Obermeier et al., Oncogene 2016).*

Thanks for pointing this out. We are aware of the report, although it appeared to us that only some of the cell lines showed increased SSA. We have now added a mention and a citation acknowledging this study in our revised manuscript. At the same time, we would like to note the much more profound increase in SSA following PALB2 knockdown and the emphasis of our study on the BRCA1-PALB2 and PALB2-BRCA2 interactions and complex formations in suppressing SSA.

*Also the analysis of the effects of a series of BRCA1 deletion mutants on HR, SSA and the response to olaparib or cisplatin is an extension of previous work (Ransburgh et al., Cancer Research 2010).*

While Ransburgh et al. did generate 4 large BRCA1 deletion mutants and measured their effects on HR, the study did not measure any effects of the deletions on SSA nor drug sensitivity. Our study provides a detailed mapping of regions of BRCA1 involved in HR, SSA and therapy response. As expected, deletion of the N-terminal RING domain, the PALB2-binding coiled-coil domain and the C-terminal BRCT domain, sensitizes cells to both cisplatin and olaparib. Moreover, our series also includes Δ486-627, which contains the NLS sequence of BRCA1 but is unable to effectively support drug resistance despite retaining ~70% HR activity, as well as Δ1056-1307 with substantially reduced HR activity despite the region having no known domains or interacting proteins identified to date. More interestingly, deletion of amino acid 887-1061 does not significantly affect HR or SSA, yet it leads to a greatly enhanced olaparib resistance. We believe that the findings from our deletion series will spawn new ideas and thinking that may significantly advance the BRCA1 field in the future.

*The data in Figure 4 confirm previous findings by Rodriguez et al. (Exp Cell Res 2004) and nicely show that the defect of BRCT mutants in HR is not primarily caused by mislocalization of overexpressed mutant BRCA1. It is interesting to speculate why BRCT mutants do not seem to enter the nucleus efficiently. However, as this has been published before, Figure 4 could be moved to supplemental data.*

We understand the concern. However, we still wish to keep the figure as a formal one due to the following reasons. First, all mutations studied in the Rodriguez paper are BRCT truncating mutations, whereas the present manuscript is focused on point mutations, which in our view are different. A strong indication is that several point mutants (A1708E and M1775R, etc.) appears to be grossly destabilized, whereas truncating mutants, such as 5055delG, are not (although frameshift mutations often lead to nonsense mediated decay (NMD) of mRNA in vivo). Also, the missense mutants may still bind known interaction partners with low affinity or even certain unknown factors that may affect their localization. Second, our Figure 4 includes not only the BRCT mutants but also mutants that affect BARD1 and PALB2 binding (C61G and M1411T, respectively); therefore, the figure serves a broader purpose and emphasizes the comparison between the different class of mutants. Third, Rodriguez et al. did not test whether BARD1 can rescue the HR defect of the BRCT mutants. Fourth, we believe we have already properly acknowledged their findings in our manuscript.

*2) Specificity and sensitivity of the assays*

*Functional analysis of BRCA1 VUS is important to aid clinical decision-making. RNAi of endogenous BRCA1 and transient overexpression of cDNA constructs may not capture all aspects of BRCA1 biology, but when the specificity and sensitivity of the assays are high, the functional data are relevant for genetic counseling. In addition to the controls mentioned by the authors, there are a number of variants known to be pathogenic or neutral (the BIC database needs to be updated). The ENIGMA consortium has published lists of these variants in Millot et al. (Hum Mut 2012). The authors should indicate all known controls and use these to calculate the specificity and sensitivity of their assays.*

Thanks for raising this issue and providing valuable advice. We have assessed the sensitivity and specificity of our HR and drug sensitivity assays using the Receiver Operating Characteristics (ROC) analysis to simultaneously capture the maximum sensitivity and specificity of the assays with 95% confidence interval as previously described (Guidugli et al., 2013), using all known controls as suggested. The result confirms that our assays are both highly sensitive and highly specific to distinguish between neutral and pathogenic variants. Threshold lines have added to the figures and the Results and Methods sections have been updated as well.

*3) Truncated BRCA1 in MDA-MB 436 cells*

*The authors used MDA-MB 436 cells to analyze the abilities of BRCA1 variants to confer cisplatin or olaparib resistance. It should be noted that MDA-MB 436 cells have a BRCT truncating mutation and express a near full-length BRCA1 protein that can be stabilized and confer cisplatin and PARPi resistance (Johnson et al., PNAS 2013). It is therefore not a BRCA1-null cell line and results have to be evaluated with caution. The authors should include a paragraph on the limitations of their assays systems in the discussion, and cite the PNAS paper from Johnson et al.*

Work by Johnson et al. is indeed helpful in the understanding of drug resistance mechanisms of BRCA1-deficient cells, and this citation has been added. At the same time, we would like to note the major differences between the two studies. First, the BRCA1 stabilization and PARPi resistance observed by Johnson et al. was after repeated selection of MDA-MB-436 cells with rucaparib over 4-6 months, whereas in our case we used only one round of selection over 3-4 weeks. Moreover, we used double selection with both G418 (to select for transfected cells) and olaparib or cisplatin (for repair function). Under the condition used, we never had a single olaparib-resistant colony in vector transfected cells or untransfected cells, which allow us to rule out any contribution of endogenous BRCA1 to our readout. On one occasion we did get 4 cisplatin-resistant colonies from vector transfected cells (as compared with roughly 100 colonies we got from wt BRCA1 cDNA), which we think could be due to the effective dose of cisplatin in the particular plate being slightly lower than the threshold, as cisplatin is not a very stable compound. Thus, we believe that MDA-MB-436 is a suitable cell line for BRCA1 functional analysis under the condition used. A brief discussion of these has been added.

*4) BRCA1 and PALPB2 protein expression levels*

*The assays rely on artificial overexpression of BRCA1 variants, which may affect the results. How does the expressed BRCA1 protein level compare with the endogenous level? Preferably, BRCA1 needs to be expressed at equal and physiologically relevant levels in the relevant cell lines and assays. If it is heavily overexpressed it could be swamping the system, e.g. diluting interacting proteins, which does become relevant later in the paper. To address this issue, the authors should show expression levels of endogenous and overexpressed BRCA1 proteins.*

Thanks for the suggestion. We have added a small panel showing the expression levels of endogenous and exogenous BRCA1 in the HR reporter cells (new Figure 1). The averaged expression level of BRCA1 in the well of cells first depleted of the endogenous protein and then transfected with wt BRCA1 cDNA was typically 3-5 times that of the endogenous level. However, the real situation is more complex, as all the cells were not transfected and the cells that were transfected would express BRCA1 at different levels, as can be seen in Figure 4. Based on our experience gained from the 6-plus years working on this project, we would say that some of transfected cells express BRCA1 very highly and these cells have a large contribution to the averaged BRCA1 level; however, there are always a few percent of (the total population of) cells that express BRCA1 at levels comparable to the endogenous one, and these are likely the cells that turn green. With these said, more important is that all VUSs but the BRCT point mutants showed expression levels similar to that of the wt protein as assayed by western, and all but the RING and BRCT mutants showed similar expression patterns when analyzed by immunofluorescence. Therefore, despite the overexpression of exogenous BRCA1, we believe that the results of our analyses are reliable. A brief mention of the above has been added to the Results section. The expression levels of the BRCT mutants are now shown in Figure 1—figure supplement 2.

*How does the expressed PALB2 protein level compare with the endogenous level (Figure 2)?*

We have added a panel of western blots to address this concern (new Figure 2). The apparent overall level of exogenous PALB2 expression is about 2-5 times that of the endogenous level, and the same caveat as noted above also applies here. For reasons unclear, the BRCA1 binding mutant L21A showed substantially higher expression than the wt protein (yet it still failed to support HR and suppress SSA).

*5) Variable knockdown efficiencies and potential off-target effects of siRNAs*

*For the assays in U2OS cells, it is essential that there is efficient knockdown of endogenous BRCA1 in all cases. Knockdown efficiency should therefore be shown.*

The knockdown efficiencies of BRCA1 and PALB2 are now shown in the above-noted new panels addressing the expression levels of endogenous and exogenous BRCA1 and PALB2 (Figure 1 and Figure 2, respectively). Please note that for our HR and SSA assays involving “protein replacement”, the knockdown was conducted in 10 cm dishes and the cells were then mixed and reseeded into 6-well plates for the second transfection, therefore the wells have uniform levels of knockdown.

*Figure 2 shows that knockdown of BRCA2 with 2 different siRNAs gives very different quantitative results: a 30-fold increase in SSA for siRNA 1949 and an 8-fold increase for the 9025. Even the PALB2 siRNAs give very high SSA increases. RAD51 protein levels are frequently affected by siRNAs (as shown by Elledge and colleagues), so it is important to show that RAD51 levels are not affected by the various siRNAs, especially 1949, which appears to be especially effective at increasing SSA while not being demonstrably better at knocking down BRCA2. Related to this, the less effective siRNA for PALB2 seems to be more effective at increasing SSA.*

Thanks for raising this issue. We have now repeated the SSA experiment using 4 more (6 total) BRCA2 siRNAs (updated Figure 2). Out of the 6 BRCA2 siRNAs, 5 of them increased SSA by 2-3 fold, whereas siRNA 1949 remained an outlier, elevating SSA by about 5 fold. The 2 PALB2 siRNAs behaved more similarly, stimulating SSA by about 6 fold. Our old results were obtained in 2013-2014 and the much larger increases in SSA following BRCA2 and PALB2 knockdown were reproducible at the time (at least 3 independent experiments). Although we cannot explain exactly why our new results show substantially less dramatic increase, we did notice time and times again that the performance of the reporter cell line, in terms of the number of green cells yielded, tend to deteriorate over time and can fluctuate greatly between consecutive passages. We also noticed that BRCA2 in the new stock of cells used for revision is not degraded as quickly upon PALB2 loss as in the old cells. The exact mechanism as to how BRCA2 and PALB2 suppress SSA awaits further investigation.

The western blotting results in the original Figure 2 have been replaced with new results that now include RAD51 (new Figure 2). All 6 BRCA2 siRNAs were found to reduce RAD51 protein abundance; however, there was no correlation between the reductions in RAD51 amount with the increases in SSA activity. Moreover, siRNA 1949 was among the ones that had the mildest effect on RAD51 level. Therefore, the reduction in RAD51 amount is unlikely to be a major cause of increased SSA. Given that one molecule of BRCA2 can bind up to 6 molecules of RAD51 (Jensen et al., 2010), the reduction of RAD51 could be explained by the co-degradation of RAD51 with their carrier BRCA2 during the siRNA treatment period; or, it could be caused by an indirect effect resulting from altered cell cycle due to loss of BRCA2 combined with potential off-target effect on other genes. In the case of PALB2 siRNAs, neither of the 2 used significantly reduced RAD51 amount; in fact, RAD51 amount even increased slightly in the case of siRNA 1493, which caused a 7 fold increase in SSA. It is therefore safe to say that changes in RAD51 abundance has limited, if any, impact on SSA in the case of PALB2 knockdown.

As to the last point, we have used the 2 PALB2 siRNAs in multiple cell lines and numerous experiments, and their knockdown efficiencies were always similar. In fact, the levels of PALB2 knockdown were also similar in our original figure (the difference was in BRCA2, for reasons unknown). Due to the use of more BRCA2 siRNAs as described above, the western blots in the original figure have been replaced with new ones, in which BRCA2 amounts were similar, as mentioned above.

*Although PALB2 knockdown results in more than 10-fold increase in SSA activity (Figure 2), re-expression of PALB2 only suppresses SSA less than 50% (Figure 2). What is the reason for this discrepancy?*

This is indeed puzzling. We have repeated the experiment several more times and the suppression was even slightly less than before (but still highly significant) (new Figure 2). In our protocol, PALB2 siRNA was first used to deplete the endogenous protein, cDNAs for PALB2 and I-SceI were delivered 48 hr later, and cells were harvested and green cells counted another ~52 hr later. It is conceivable that the depletion of PALB2 not only caused loss of the PALB2 itself but also led to certain secondary effects as a result of PALB2 loss, such as potential changes in chromatin state, that cannot not be immediately restored upon sudden re-expression of PALB2. It is also possible that in order for PALB2 to function in SSA suppression, it needs to be post-translationally modified or form functional complexes with other proteins, which cannot be fully achieved immediately after re-expression.

*6) Statistical analysis*

*At present, it is unclear which of the observed differences are significant. Statistical analysis is lacking as are source data files and both should be submitted.*

We have added p values for relevant parts of Figure 2 and Figure 5. For the larger sets of HR and drug resistance data, providing a p value to each variant is unnecessary or even counterproductive, as some of the small differences will be significant due to the number of repeats we have conducted, even though the difference could in fact be simply due to a slight variation of plasmid quality or other technical issues. For these data, a cutoff threshold is provided, with p values added only to M1400V. Source data are now submitted.